# Characterization of the Survival Influential Genes in Carcinogenesis

**DOI:** 10.3390/ijms22094384

**Published:** 2021-04-22

**Authors:** Divya Sahu, Yu-Lin Chang, Yin-Chen Lin, Chen-Ching Lin

**Affiliations:** Institute of Biomedical Informatics, National Yang Ming Chiao Tung University, Taipei 11221, Taiwan; sahu.divya786@gmail.com (D.S.); skyblue8863@gmail.com (Y.-L.C.); ying87000@gmail.com (Y.-C.L.)

**Keywords:** survival influential genes, pan-cancer, prognostic biomarkers

## Abstract

The genes influencing cancer patient mortality have been studied by survival analysis for many years. However, most studies utilized them only to support their findings associated with patient prognosis: their roles in carcinogenesis have not yet been revealed. Herein, we applied an in silico approach, integrating the Cox regression model with effect size estimated by the Monte Carlo algorithm, to screen survival-influential genes in more than 6000 tumor samples across 16 cancer types. We observed that the survival-influential genes had cancer-dependent properties. Moreover, the functional modules formed by the harmful genes were consistently associated with cell cycle in 12 out of the 16 cancer types and pan-cancer, showing that dysregulation of the cell cycle could harm patient prognosis in cancer. The functional modules formed by the protective genes are more diverse in cancers; the most prevalent functions are relevant for immune response, implying that patients with different cancer types might develop different mechanisms against carcinogenesis. We also identified a harmful set of 10 genes, with potential as prognostic biomarkers in pan-cancer. Briefly, our results demonstrated that the survival-influential genes could reveal underlying mechanisms in carcinogenesis and might provide clues for developing therapeutic targets for cancers.

## 1. Introduction

Genes with an impact on the survival of tumor cells [1,2] likely influence the survival of cancer patients. So far, cancer-essential genes have been discovered by the clustered regularly interspaced short palindromic repeats (CRISPR) method [3] and the cancer-dependent genes with the Cancer Dependency Map (DepMap) [4]. The discovered cancer-essential genes could facilitate the development of promising cancer therapies [5] and carcinogenesis mechanisms. Although experimental studies hold promise for the accurate detection of essential genes, they are capital and labor intensive, and time consuming. Therefore, several computational approaches have been implemented to predict essential genes in *Saccharomyces cerevisiae*, *Escherichia coli* [6], and humans [7,8]. However, the identification of survival-influential genes (SIGs) in cancer patients and the investigation of their role in carcinogenesis are largely unexplored.

Survival analysis is a branch of statistics that analyzes data, where the outcome variable is the time until one or more events occur, e.g., death [9]. That is, the methods of survival analysis measure the proportion of a population that will survive to passing a certain time point, i.e., the survival rate of the population. Similarly, the experimental approaches identifying essential genes measure the growth curves of the studied organism population, such as yeast or *E*. *coli* [10,11,12,13,14]. However, it is impossible to recruit humans as experimental targets with the purpose of screening essential genes or synthetic lethal gene pairs in this way. Recently, benefiting from the large projects collecting information on cancer patients’ genomics and transcriptomics data, such as TCGA (The Cancer Genome Atlas), many patients’ clinical information, including survival time, was also gathered. These sets of clinical data make survival analysis a promising strategy for recovering SIGs in cancer patients.

To identify and analyze the SIGs in cancer patients, we designed an in silico approach, incorporating effect size derived from the Monte Carlo algorithm into the Cox regression model, to assess the influence of gene expression on patient survival in cancer. We demonstrated that the pan-cancer SIGs identified by our approach are significantly associated with known oncogenic genes, and the SIGs identified in the individual cancer types displayed cancer-dependent properties. Furthermore, we found that the harmful SIGs might be involved in cell proliferation to promote carcinogenesis. However, the diverse functions of protective SIGs across cancer types demonstrate that the strategies against carcinogenesis developed by patients with different cancer types might be variable. Here, we present list of SIGs which could reveal the underlying mechanisms in carcinogenesis and propose a set of prognostic biomarkers that might be potential therapeutic targets for cancer.

## 2. Results

### 2.1. Overview of the Identified Survival Influential Genes in Cancers

In this study, we proposed a cancer-dependent approach that incorporated the Monte Carlo algorithm into Cox regression to identify the SIGs in cancers. Briefly, this approach created the simulated gene expression profiles of cancers through the Monte Carlo processes. Then, by comparing with the simulated expression profiles, this approach can estimate the effect size of the survival influence for genes in a certain cancer type. We then calculated and investigated the effect sizes for all genes in every cancer separately, to systematically define the proper threshold to identify SIGs for each cancer. We observed that as the *p*-value increased, the effect size increased at first, and then decreased afterward (Figure 1A,B, left panel). This result indicated the existence of a global maximal effect size. Furthermore, we observed that the *p*-values reaching the maximal effect size were distinct among cancer types (Figure 1A,B, right panel). This observation confirmed that a distinct threshold of significance level is necessary for different cancer types when performing a survival analysis [15,16]. Accordingly, in different cancer types, we used a *p*-value reaching the maximum effect size as the threshold to identify the SIGs (Figure 1B, right panel). The identified SIGs are listed in Appendix A. We observed that, among the 16 cancer types, eight cancer types included more harmful genes and the others included more protective genes (Figure 1C). Additionally, the proportion of SIGs ranged from 5% to 11%, except for low grade glioma (LGG) and kidney renal clear cell carcinoma (KIRC) with 31% and 20%, respectively (Figure 1C). The high proportion of SIGs in LGG and KIRC implied that the mortality of the patients with LGG or KIRC might be more sensitive to aberrant gene expression. Similarly, a previous study reported a significant association of ribonucleoproteins with poor patient survival in KIRC, LGG, and kidney renal papillary cell carcinoma (KIRP) [17]. Additionally, we performed the proposed approach to detect SIGs in pan-cancer. We finally identified 2450 pan-cancer SIGs: 1196 harmful and 1254 protective genes.

Next, to assess the association between the identified SIGs and cancer, we complied a cancer-associated gene list from three datasets. Since the data on tumor suppressor genes are limited, this cancer-associated gene list mainly consisted of oncogenes, i.e., cancer essential or cancer dependent genes. In other words, they are potentially harmful to patient survival. On the other hand, the three datasets collected these cancer-associated genes from various cancer types. Therefore, we used them to evaluate the cancer association of the pan-cancer harmful SIGs. Additionally, we used these cancer-associated genes to assess the performance of our approach in the identification of SIGs. We also compared our approach with the conventional method with two fixed thresholds, which were *p*-value < 0.05 and <0.01, and the Cox regression model with the least absolute shrinkage and selection operator (LASSO). We observed that our approach identified with a higher precision the proportion of cancer-associated genes than these three approaches (Figure 1D). Furthermore, compared with the small but highly confident harmful gene set collected from the literature, the precision of our approach was lower but comparable (Figure 1D). It is worth noting that the complied cancer-associated genes were significantly enriched in the identified pan-cancer harmful genes, but not significantly enriched in the literature-curated gene set (Figure 1D). This observation might suggest that our pan-cancer harmful genes are more associated with the compiled cancer-associated genes than the literature-curate ones. Briefly, these comparisons further demonstrated the better performance of our approach, and might show the confidence of the identified SIGs.

### 2.2. Exclusivity of the SIGs and Identification of the Pan-Cancer SIGs

To understand the characteristics of SIGs, we first investigated their prevalence across cancers. We observed that 5684 genes were identified as survival influential in only one cancer type (Figure 2A, unique region), and 9327 genes in at least two cancer types (Figure 2A, shared region). Moreover, around 90% of harmful and protective genes are in less than two cancer types, and no genes are survival influential in more than eight cancer types (Appendix A). This observation indicates that the SIGs might be exclusive between cancers. However, a too strict cutoff for identifying the SIGs might cause exclusivity. To examine this scenario, we removed the condition of significance level and only used the hazard ratio to determine the survival risk of the tested genes. That is, we denoted one gene with a hazard ratio >1 and ≤1 as harmful and protective, respectively. Accordingly, each gene was classified as either harmful or protective. Interestingly, we observed that only six genes were identified as harmful in all sixteen cancer types and no genes were protective in all 16 cancer types. Moreover, only 4.66% and 4.55% of genes were identified as harmful and protective, respectively, in more than 13 cancer types (Figure 2B); and 45.03% of genes had a mixed influence (harmful vs. protective: 8:8, 7:9, or 9:7) to patient survival across the 16 cancer types. In other words, even though we removed the significance cutoff of SIGs, the survival risk for a gene was still barely consistent across cancers. Accordingly, we concluded that the exclusivity of SIGs across cancers was not biased by the cutoff significance. Furthermore, these above results also demonstrated that our approach might be able to identify the particular SIGs in each cancer type.

On the other hand, the above observations unveiled that the genes identified as harmful or protective in a large number of cancer types might possess the potential to be pan-cancer SIGs. Indeed, genes with a more consistent pattern of survival influence across multiple cancers tended to possess a higher survival influence (*z*-score) in the merged cancer data set (Figure 2C). In other words, the identified pan-cancer SIGs tended to keep a consistent pattern of survival influence across cancers. However, the pan-cancer harmful and protective genes were not observed to affect patient survival significantly in the individual cancer types, though they possessed stronger effects than other non-SIGs (also excluding pan-cancer SIGs) in most cancer types (Appendix A). This observation showed that the survival influence of pan-cancer harmful and protective SIGs was amplified and recognized in the merged cancer data set, but was easy to ignore when they were considered for each cancer type. More importantly, without using the merged cancer data set, these pan-cancer SIGs could be omitted.

### 2.3. Analysis of SIG Roles in the Human Co-Expressed Protein Interaction Network

Previous studies have demonstrated that the investigation of genes/proteins in the biological network is powerful for elucidating their roles in the molecular mechanisms during cancer development [18,19]. We investigated the properties of SIGs in the biological system through studying their interactions in the human protein interaction network (PIN), which was obtained from InBio Map [20]. Notably, the PIN used here was static and therefore might not provide conditional information, e.g., cancer-activated protein-protein interactions (PPIs). We further utilized the co-expressed PPIs (CePPIs) to extract the context-dependent PPIs in each cancer type. In this study, we denoted a PPI formed by two genes significantly co-expressed with each other (z-score of Spearman correlation coefficient ≥2) as a CePPI. We observed that the PPIs within harmful or protective SIGs showed a significantly higher coherent expression pattern than other PPIs in the PIN (Figure 3A and Appendix A), demonstrating the stronger functional association among SIGs with similar survival influence. On the contrary, PPIs between harmful and protective genes tended to exhibit a negative correlation of expression pattern (Figure 3A and Appendix A). This observation might suggest an antagonistic effect between harmful and protective genes in cancer patients.

The proteins encoded by cancer genes have been observed to occupy more pivotal positions, e.g., hubs, bottlenecks, or the center, than other proteins in the human interactome [18]. Indeed, we found that in ten cancer types the SIGs, either harmful or protective, had a significantly higher degree of co-expression than the non-SIGs (Figure 3B and Appendix A), that is, they tended to be hubs in the cancer CePINs. Additionally, we found that, in the corresponding cancer type, except for OV, when one type of SIG showed a higher co-expression degree than non-SIGs, the other type showed a lower (Figure 3B). This competition of hub positions further supported the antagonistic effect between harmful and protective genes in cancer patients. Moreover, in the static human PIN, which does not filter out non-co-expressed PPIs, we did not observe a significant difference of degree between the SIGs and non-SIGs in most cancer types (Appendix A). Therefore, the SIGs identified in the individual cancer type might be cancer-specific instead of general hubs. These results also confirmed that our approach could identify the context-sensitive SIGs, and cancer-specific, for cancers.

### 2.4. The Survival Influential Functional Modules

Next, we performed a network-based functional enrichment analysis to identify the survival-influential functional modules, which are the protein interacting functional modules formed by the coherently expressed SIGs in the corresponding cancer type. However, we did not identify harmful modules in ESCA, HNSC, and OV or protective modules in ESCA and PAAD under the predefined criterion. Through this analysis, we could pinpoint the molecular mechanisms by which the SIGs are involved in carcinogenesis affect patient prognosis. In the harmful SIGs, we observed that the functional modules associated with cell cycle were consistent across six cancer types: BRCA, KIRC, LGG, LIHC, LUAD, and PAAD (Figure 4). Interestingly, except for CESC, other cell cycle-related modules also appeared in the remaining cancer types: positive regulation of cell proliferation in COAD and GBM; positive regulation of cell differentiation in BLCA, LUSC, and STAD; and G1/S transition of mitotic cell cycle in SARC. Briefly, twelve cancer types presented the harmful modules involved in cell cycle, implying that the dysregulation of cell cycle could be harmful to patients across cancers [21,22]. Indeed, we found that, in the merged cancer dataset, the harmful SIGs were also significantly overrepresented in the functional modules associated with cell cycle, such as “mitotic nuclear division”, “cell cycle phase transition”, and “mitotic cell cycle process” (Figure 4). This observation also demonstrates that the harmful SIGs between different cancers might be exclusive, but the biological processes of harmful SIGs (regulation of cell cycle) might demonstrate the generality of harmful SIGs’ survival characteristics across cancers. However, the functional modules from the protective SIGs were more diverse than the harmful modules. The most prevalent modules, which were discovered in CESC, HNSC, LUAD, and SARC, are relevant for immune response (Figure 5). The activation of immune response-related functions is known to improve patient prognosis [23,24]. The functional modules of protective SIGs identified in the merged cancer dataset are involved in the regulation of gene expression and protein trafficking (Figure 5). This is inconsistent with the modules identified from the individual cancer types. This observation may confirm the exclusivity of protective SIGs between cancers, and imply that patients with different cancer types might develop different mechanisms against carcinogenesis.

### 2.5. The Cancer Hallmarks of SIGs in Pan-Cancer

To investigate how these SIGs influence carcinogenesis in pan-cancer, we obtained hallmark gene sets from the Molecular Signature Database (MSigDB) [25,26,27] and performed enrichment analysis using Fisher’s exact test. These hallmark gene sets are well-defined and representative biological processes and pathways in cells. We observed that pan-cancer harmful genes were significantly over-represented for all the hallmark categories (Figure 6A). This observation implies that the way harmful SIGs are involved in carcinogenesis might be variable and prevalent in respect to cellular function. More specifically, we found that the pan-cancer harmful SIGs were significantly enriched with several cancer-relevant hallmarks, such as epithelial mesenchymal transition, DNA repair, glycolysis, hypoxia, apoptosis, MYC target, p53 pathway, and E2F targets (see full significant list in Appendix A) [28,29]. This result suggests that the pan-cancer harmful SIGs might promote carcinogenesis through participating in these cancer hallmark processes. In contrast, we observed that pan-cancer protective genes were significantly overrepresented only in the metabolic category, but underrepresented in the immune response and pathway categories (Figure 6B). Interestingly, two cancer-relevant hallmarks, bile acid and fatty acid metabolism, in which pan-cancer protective SIGs are enriched have been reported to contain cancer-therapeutic potential [30,31].

To investigate the pan-cancer harmful and protective SIGs in hallmark categories associated with patient survival, we built a risk score using the univariate z-score for overall survival for each patient. We observed that pan-cancer harmful and protective SIGs in the MSigDB category tended to have a positive and negative standardized risk score, respectively (Figure 6C,D). This observation suggested that the pan-cancer harmful genes in all the MSigDB categories were significantly associated with poor patient survival, emphasizing the role of the pan-cancer harmful SIGs in promoting carcinogenesis through involving the cancer-relevant hallmarks. In contrast, the pan-cancer protective genes in the signaling and metabolic category were highly significantly associated with good patient survival. Collectively, these observations further highlighted the potential roles played by the identified pan-cancer SIGs in carcinogenesis.

### 2.6. Identification of Clinically Relevant Pan-Cancer Harmful SIGs in the Proliferation Hallmark

According to the above results, we could conclude that the harmful genes involved in proliferation are critical to the poor prognosis of cancer patients. Therefore, we selected pan-cancer harmful genes in the proliferation category and performed a series of analyses. First, we performed a pairwise co-expression analysis by applying spearman correlation, and then transformed the obtained SCCs into z-scores using Fisher’s z-transformation in the merged cancer dataset, and also in each cancer. With a z-score cut-off of three, we identified 5950 co-expressed gene pairs in common, suggesting a strong correlation between harmful genes in all 16 cancer types. To have a more detailed view of these co-expressed genes, we identified 174 of them possessing a unique protein-protein interaction among the 96 pan-cancer harmful genes (harmful proteins) in the human interactome. Highly connected nodes are usually defined as “hubs”. We filtered the 10 genes (*CDK1*, *CDC20*, *PLK1*, *AURKA*, *AURKB*, *BRCA1*, *BUB1B*, *MCM2*, *BUB1*, and *MCM7*) which were the top 10 highly connected and having a maximum degree of 20 and a minimum degree of 7 (Figure 7A). In addition, two distinct sub-networks were observed. The large sub-network consisted of key cancer proliferation genes mainly categorized as kinases, while the other small sub-network were mainly categorized as proteasomes (Figure 7A). We also identified that these genes were differentially expressed in primary tumor and matched normal samples in pan-cancer (Figure 7B) and also upregulation in 10 cancer types (Appendix A), further suggesting the oncogenicity of these pan-cancer harmful SIGs. We further categorized the expression profile of these 10 highly co-expressed genes into low expression groups and high expression groups, based on their median expression. We found the high expression levels of these genes were significantly associated with poor survival of patients through pan-cancer analyses (Figure 7C–L). We also found that some of the 10 genes showed clinical relevance by associating with patient survival in at least 11 cancer types (Appendix A). Interestingly, the Kaplan-Meier plot of *AURKA*, *AURKB*, *BUB1*, *BUB1B*, *CDC20*, *CDK1*, and *PLK1* showed a wide clear separation between low expression and high expression curves in KIRC and LGG (Appendix A), suggesting them as prognostic biomarkers. For example, *CDC20*, a potential novel target for cancer therapy has been reported to be dysregulated in the majority of human cancers, including oral squamous cell carcinoma [32], human bladder carcinoma [33], pancreatic cancer [34], colorectal cancer [35], breast cancer [36], glioblastoma [37], and non-small cell lung cancer [38], and showed clinical relevance in six cancer types, i.e., BLCA, KIRC, LIHC, LUAD, LGG, and PAAD (Appendix A), while *PLK1*, another potential target for cancer therapy [39] showed clinical relevance in eight cancer types, including BLCA, COAD, HNSC, KIRC, LIHC, LUAD, LGG, and PAAD (Appendix A). Taken together, a strong correlation between these ten SIGs and their significant association with major cancer types further suggested their potential role in promoting carcinogenesis.

## 3. Discussion

In this study, we identified genes influencing patient survival in 16 individual cancer types and pan-cancer. Moreover, our approach tried to identify the survival influential genes depending on each cancer type, instead of using a global and fixed cut-off across all the studied cancer types. Intuitively, the survival influential genes should not be prevalent in the genome to control the fatal sensitivity to the perturbations inside or outside the cells. Accordingly, previous studies have reported that the proportion of essential genes in the human cancer genome is only about 10% [1,40,41]. Likely, the proportion of SIGs identified by our approach was around 10%, except for LGG and KIRC, which were still with a relatively higher proportion of around 31% and 20%, respectively (Figure 1C). This observation might reveal a limitation of our approach. Even so, our approach outperformed the conventional method of a fixed threshold and LASSO algorithm, and successfully identified the SIGs possessing cancer-dependent properties from co-expressed network analysis. However, the permutation strategy is computationally expensive, thus a more effective approach for assessing cancer-dependent thresholds to determine the survival-influential genes in different cancers is needed.

Furthermore, we observed that the expression level of survival-influential genes in the corresponding cancer type is distinguishable from the other cancers (Appendix A). Interestingly, in the corresponding cancer type, the harmful SIGs were down-regulated, and the protective SIGs were up-regulated, compared to other cancer types (Appendix A). This observation might suggest one imaginable scenario: cancer patients fight against carcinogenesis through inhibiting the harmful SIGs and/or activating the protective SIGs. It is worth noting that most samples in the data set are biased to alive patients; namely the harmful and protective genes could be repressed and promoted respectively to aid patient survival. Indeed, we observed that, in deceased patients, the expression level of harmful and protective genes was elevated and decreased, respectively (Appendix A). However, this observation could be biased by the objective of the Cox regression model, to assess the association between covariates (gene expression level in this study), and the risk of patient survival. That is to say, genes increasingly or decreasingly expressed in deceased samples compared to censored ones have a higher probability of being identified as harmful or protective genes, respectively. Indeed, the *z* scores of coefficients from the Cox regression model were significantly and positively correlated with the Cohen’s D of gene expression level between deceased and censored samples (deceased-censored) for all 16 cancer types (the averaged Spearman correlation coefficient was 0.6, all *p*-values were smaller than 10^−128^). Interestingly, we found that the harmful and protective genes were expressed less and more, respectively, in censored (alive) patients with early stage tumors compared to late stage in 7 out of 13 cancer types (Appendix A). This observation implies that, in the patients with a lower risk of survival, the harmful and protective genes could be repressed and promoted, respectively. However, experiments to demonstrate the effect of survival influential genes on carcinogenesis or patient survival are required to validate this observation.

Additionally, we have identified ten clinically relevant genes whose expression can identify the high-risk group of patients with poor outcome in various cancer types. Our findings were also supported by various published literature reports. High expression of *CDC20* showed an association with tumor recurrence and patient death in bladder cancer and pancreatic cancer [33,34]. Additionally, high expression of *CDC20* was significantly associated with overall survival in advanced clinical stage (stage III and IV) patients with colorectal cancer [35]. A meta-analysis report from five studies on approximately 700 colorectal patients showed that high expression of *PLK1* was associated with worse patient survival [42]. Moreover, multivariate cox regression analysis, after adjusting for clinicopathological factors confirmed that high PLK1 expression at the protein level was independently associated with poor outcome in patients with lung squamous cell carcinoma [43].

## 4. Materials and Methods

### 4.1. Data Collection and Preprocessing

The RNA-Seq V2 transcriptome expression datasets and corresponding clinical information from The Cancer Genome Atlas (TCGA) were downloaded from the UCSC Xena browser [44]. We used normalized read counts inferred via RSEM (RNA-Seq by Expectation Maximization) algorithm from RNA-Seq V2 data as mRNA expressions. To increase the statistical power of the survival analysis, we only included the cancer types with sufficient samples (>150) and deceased events (>50); and kept genes and samples with less than 50% unexpressed (RSEM = 0) samples and genes, respectively. Finally, we obtained 6584 samples in total (Appendix A) and around 17,690 detectable genes on average (Appendix A) across 16 cancer types (Appendix A). To identify survival influential genes in pan-cancer, we assembled a pan-cancer gene expression profile that covered 6584 samples and 16,056 intersecting genes across 16 used cancer types, and termed it merged cancer. The studies in the pan-cancer atlas from TCGA developed and adopted this strategy to enhance the pan-cancer features cross cancers [45,46,47].

### 4.2. Identification of Survival Influential Genes in Cancers and Pan-Cancer

In this study, we applied the Cox regression model as the main framework to discover survival influential genes in cancers. We first used the univariate Cox regression model to examine the association between clinical variables and patient survival time and to identify the potential clinical confounding factors (Appendix A). Then, we utilized the multivariate Cox regression model (the covariates are the expression level of the tested gene and the identified significant clinical confounding factors) to assess the influence of a gene on patient survival. In this study, we denoted the genes with positive and negative coefficients as harmful and protective to patient survival, respectively. The significance of the coefficient β was determined by comparing with the null model, which hypothesizes that the changes of tested gene expression level have no effect on patient survival, and was further tested by a likelihood ratio test and the Wald test [48,49]. The Wald test examines whether the observed regression coefficient statistically differs from zero, a reported a *z*-score (standard score), and a *p*-value estimated by *z*-score for the observed regression coefficient. The Cox regression model was performed using the *Survival* R package [50].

Previous studies have observed that the distributions of *p*-values derived from the Cox-regression model vary across cancers [15,16]. Accordingly, we applied the Monte Carlo algorithm combined with effect size to determine the *p*-value threshold of the regression coefficient for each cancer type independently. First, we created 1000 simulated cohorts for each cancer type. Each simulated cohort was provided with the same number of patients in the real data, but with randomly permutated survival information. Notably, both survival time and deceased status were coupled as a pair to be permutated. Accordingly, the simulated cohort contained identical survival profiles but randomly permutated gene expression profiles. Then, we used a Cox-regression model on each simulated cohort to assign the simulated *p*-values of regression coefficient to genes. We then calculated the odds ratio to estimate the effect size of each significance level. The odds ratio can be described as below:(1)$${OR}_{p}=\frac{O_{I}}{O_{S}}=\frac{N_{SI}}{N_{SS}}/\frac{N_{II}}{N_{IS}}$$
where *OR_p_* is the odds ratio of the significance level *p*-value < *p*; *O_I_* (*O_S_*) is the odds of significant genes in real data being identified as insignificant (significant) in the simulated data; *N_SI_* and *N_SS_* (*N_II_* and *N_IS_*) represent the number of significant (insignificant) genes in real data being identified as insignificant and significant in the simulated data, respectively. In other words, the odds ratio in this study described the disagreement between the significant genes in the real data and the insignificant genes in the simulated data under the given significance level.

For each cancer type, the above steps were repeated for 1000 simulated cohorts, and the mean odds ratio was used as the effect size of each significance level. We then applied a *p*-value reaching the maximum effect size as the threshold to determine the survival-influential genes. The effect size was calculated for harmful and protective genes separately.

We then performed the same procedure on the merged cancer data set to identify the pan-cancer SIGs. To increase the generality of the pan-cancer SIGs, we created 16 cancer minus gene expression profiles that merged gene expression profiles from 15 cancer types: for example, the BRCA minus merged the gene expression profiles from 15 out of the 16 used samples, except for BRCA. We then performed the same procedure on each cancer minus data set to identify cancer minus SIGs. Subsequently, we defined the genes identified as SIGs in (1) the merged cancer dataset, (2) at least one cancer type, and (3) all cancer minus datasets as the pan-cancer SIGs.

### 4.3. Compilation of the Cancer-Associated Genes

Next, to assess the association between the identified SIGs and cancer, we complied a cancer-associated gene list from three datasets: (1) 688 cancer genes from the COSMIC release (v85) [51]; (2) 550 cancer essential genes screened from the CRISPR system in human cancer cell lines [3]; and (3) 769 cancer genes from the cancer dependency map [4]. Additionally, we collected a small but highly confident harmful gene set from the literature [52,53,54,55,56,57]. We then performed enrichment analysis using Fisher’s exact test to evaluate the association between SIGs and cancer.

### 4.4. Functional Modules of SIGs

To investigate the biological processes of SIGs involved in carcinogenesis, we performed conventional and network-wise functional enrichment analyses to identify the functional modules. The functional annotations of genes were obtained from gene ontology (GO) [58,59]. In the conventional functional enrichment analysis, the significance was established by *p*-value derived from the hypergeometric test, which is described below:(2)$$P\left(X=k\right)=\frac{\left(\begin{array}{c} m\\ k \end{array}\right)\left(\begin{array}{c} N-m\\ n-k \end{array}\right)}{\left(\begin{array}{c} N\\ n \end{array}\right)}$$
where *X* denotes the evaluated function; *N* represents the number of GO annotated genes; *m* indicates the number of SIGs; *n* represents the number of genes with the evaluated function; and *k* indicates the number of SIGs with the evaluated function. To enhance the functional relationship between the identified genes, protein interaction and network-wise functional enrichment analyses were incorporated to discover the functional modules within SIGs [60]. The source of the human protein-protein interaction (PPI) data was InBio Map [20]. The network-wise functional enrichment analysis was modified from the conventional method. The significance of the tested function was based on *p*-values produced from a modified hypergeometric test. The hypergeometric distribution for the network-wise approach is described below:(3)$$P_{e}\left(X=k_{e}\right)=\frac{\left(\begin{array}{c} m_{e}\\ k_{e} \end{array}\right)\left(\begin{array}{c} N_{e}-m_{e}\\ n_{e}-k_{e} \end{array}\right)}{\left(\begin{array}{c} N_{e}\\ n_{e} \end{array}\right)}$$
where *e* is the abbreviation of the PPI. Each symbol has the same meaning as in the conventional hypergeometric distribution, but the counting objects are changed from genes to functional PPIs. The functional PPIs are interactions formed by the two genes involved in the same functions. This approach revealed the significant protein interaction functional modules in which the identified genes were involved. To incorporate the conditional information (related to cancer) into the functional modules, we only studied co-expressed PPI in the network-wise functional enrichment analysis; and genes that were formed by co-expressed PPI in the conventional analysis. Furthermore, we used co-expressed PIN as a background instead of the static PIN to enhance the cancer-dependent constraint. Subsequently, the function modules with adjusted *p*-value <0.05 from both analyses were identified as the significantly enriched function modules. These two *p*-values were adjusted by the Benjamini and Hochberg multiple testing procedures to control the false discovery rate (FDR) [61]. Additionally, only the significant function modules with a depth greater than six in the GO database were used to increase the functional specificity. The depth is the shortest path length from a function to the root, which is the term biological process (GO:0008150), in the GO tree structure. Finally, the significant functional modules with more than five SIGs and five co-expressed PPIs were identified as the functional modules for the following analyses.

The discovered functional modules were further summarized by the REVIGO [62] algorithm, with a similarity ≥0.9, which was calculated from the SimRel algorithm [63] and visualized using the CirGO package [64].

## 5. Conclusions

In this study, we tried to identify SIGs in cancers and pan-cancer and investigated the characteristics of SIGs in carcinogenesis. In summary, the SIGs are highly exclusive across cancers, and tend to occupy pivotal positions in the co-expressed PIN associated with the cancer. More specifically, the harmful SIGs prefer to participate in the biological processes related to cell cycle and cell proliferation. However, the functions of protective SGIs are variable across cancer types, but might be involved in bile acid or fatty acid metabolism to improve patient survival. Briefly, the identified SIGs may provide a potential biomarker pool for cancer therapy, and facilitate improving our knowledge of the molecular mechanism of carcinogenesis.

## Figures and Tables

**Figure 1 ijms-22-04384-f001:**
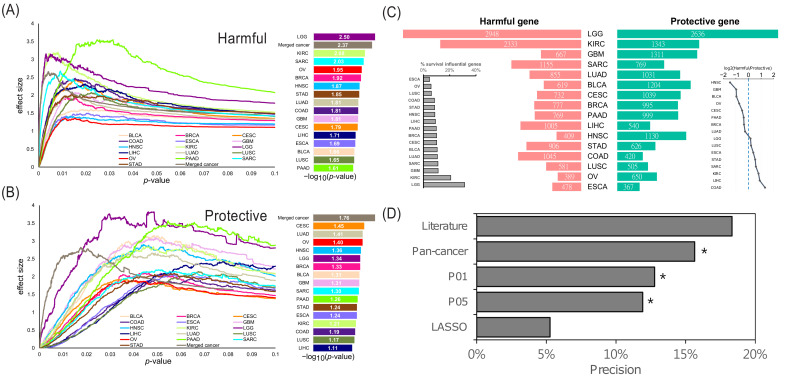
Summary of the survival-influential genes in cancers. (**A**,**B**) The effect sizes of the different threshold *p*-values for each cancer type. The effect sizes of *p*-values were estimated via 1000-time permutation test. The *p*-value producing the largest effect size (right panel) was denoted as the cut-off of significance level for identifying survival influential genes. (**C**) The number of identified survival influential genes in each cancer type. Red and green bars represent harmful and protective genes respectively. The proportion of survival influential genes and the ratio of harmful to protective genes in each cancer genome are also displayed. (**D**) Cancer association of the SIGs. Precision is the proportion of cancer-associated genes in the SIGs identified from LASSO (Cox regression model with LASSO regularization), P05 and P01 (Cox regression model with the threshold of *p*-value < 0.05 and 0.01), pan-cancer, and the literature. An asterisk signifies a significantly large proportion of caner-associated genes (*p*-value < 0.05, Fisher’s exact test) in the corresponding category.

**Figure 2 ijms-22-04384-f002:**
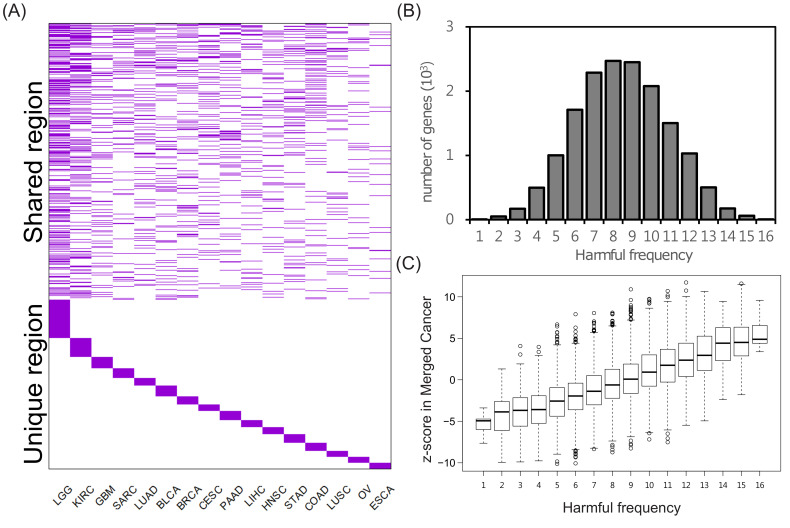
Exclusiveness and generality of SIGs. (**A**) Presence matrix of the survival influential genes across cancers. Each column indicates cancer type and each row the union of both harmful genes and protective genes of 16 cancer types. (**B**) The distribution frequency in which the genes were identified as harmful across cancers. (**C**) The association between the z-score of genes in the merged cancer data set and the frequency with which the genes were identified as harmful across cancers.

**Figure 3 ijms-22-04384-f003:**
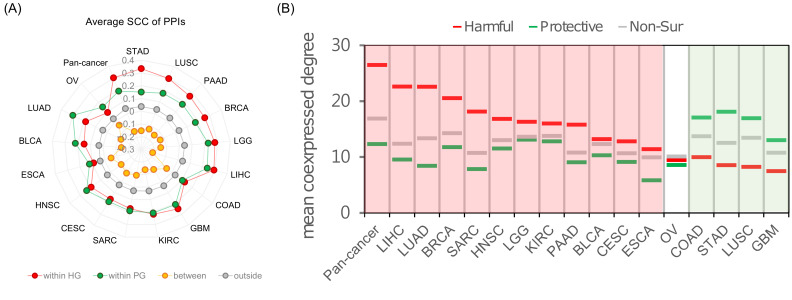
Co-expressed protein-protein interaction among SIGs. (**A**) The average Spearman correlation coefficient (SCC) between SIGs in cancers. The average SCCs were calculated between two genes forming PPI only. The red and green circles represent the mean SCCs within harmful and protective SIGs, respectively; yellow circles are the mean SCCs between harmful and protective genes, between SIGs with different types; grey circles are SCCs between non-SIGs. (**B**) Comparison of the co-expressed degree between harmful and protective SIGs and non-SIGs. The co-expressed degree is the number of significantly co-expressed (z-score of SCC ≥ 2) PPI partners of one gene.

**Figure 4 ijms-22-04384-f004:**
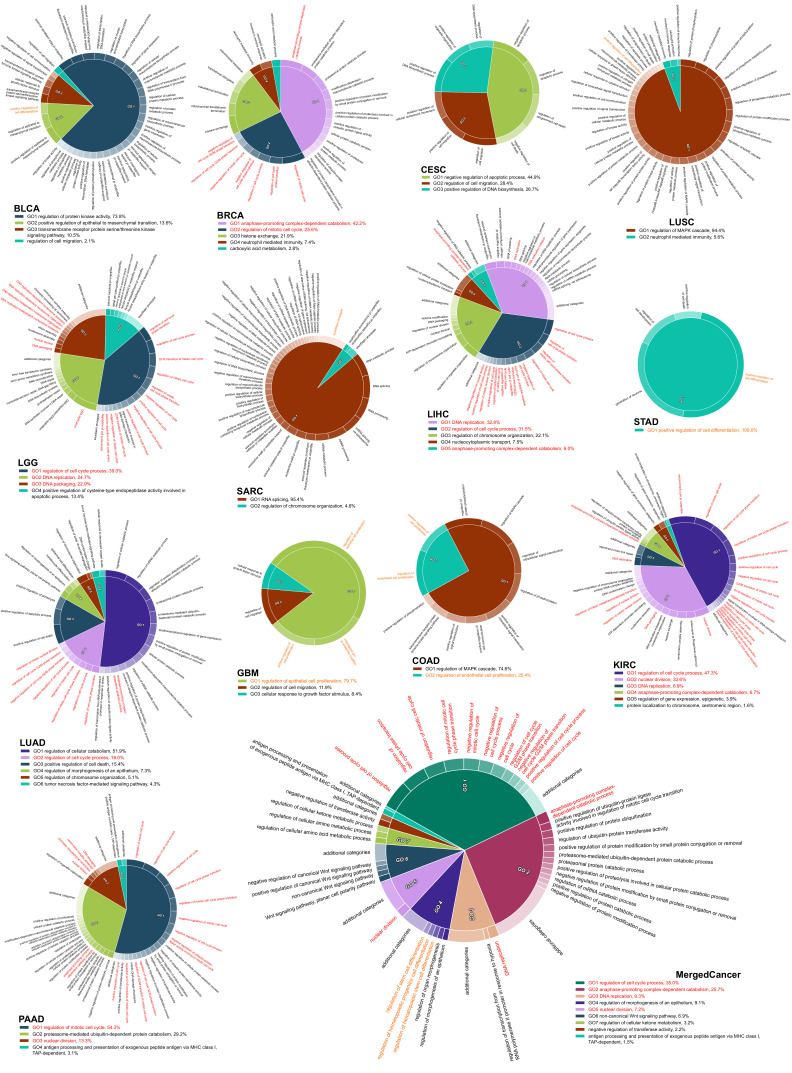
The survival influential modules formed by harmful SIGs. Each pie chart shows the identified functional modules in which the harmful SIGs are involved in the corresponding cancer type. The percentage is the relative significance of one functional module to the others. The functional modules associated with cell cycle are marked by red; modules related to cell proliferation or differentiation are marked by orange.

**Figure 5 ijms-22-04384-f005:**
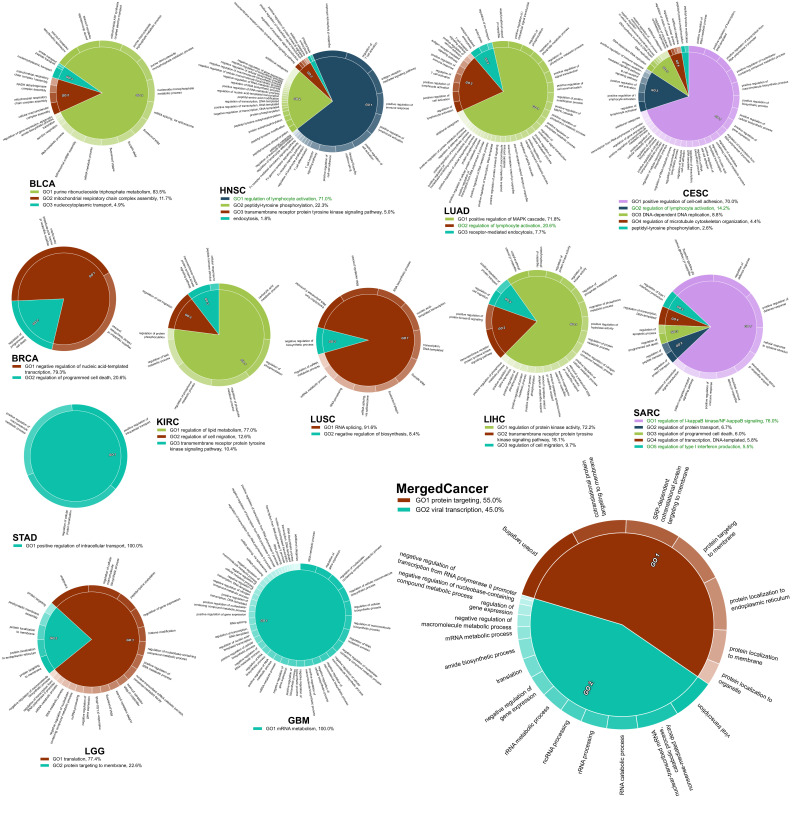
The survival influential modules formed by protective SIGs. Each pie chart shows the identified functional modules in which the protective SIGs were involved in the corresponding cancer type. The percentage is the relative significance of one functional module to the others. The functional modules associated with immune response are marked by green.

**Figure 6 ijms-22-04384-f006:**
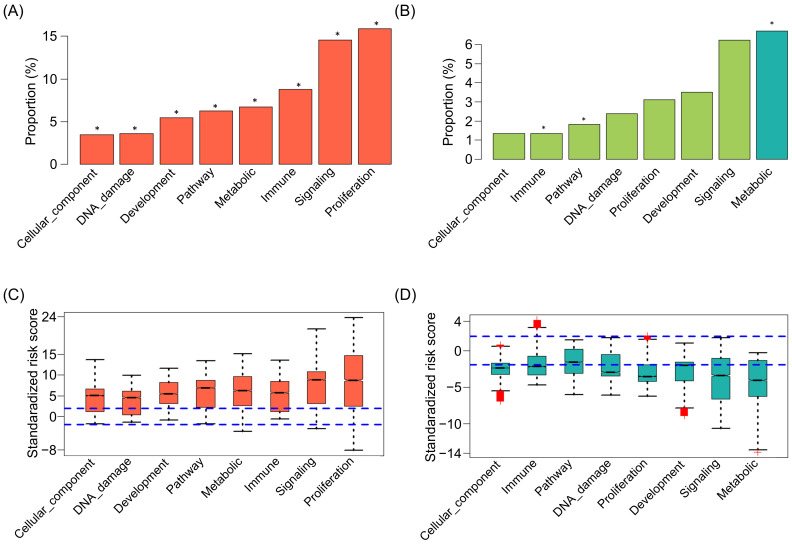
Enrichment analysis of pan-cancer SIGs in cancer-relevant hallmarks. (**A**,**B**) Overrepresentation or underrepresentation of pan-cancer harmful (**A**) and protective (**B**) genes in MsigDB hallmark gene sets. The *x*-axis shows the MSigDB hallmark category. The *y*-axis shows the proportion of the pan-cancer harmful (**B**), and protective (**C**) genes of the total genes in the MSigDB category. Dark and light bars represent overrepresentation and underrepresentation, respectively. An asterisk signifies a significant proportion of pan-cancer SIGs (*p*-value < 0.05, Fisher’s exact test) in the corresponding hallmark category. (**C**,**D**) Risk score of pan-cancer harmful (**C**), and protective (**D**), SIGs in the MSigDB category compared to overall patient survival. Positive scores represent a high expression level of genes significantly associated with poor survival, and negative scores indicate better survival. The dashed lines represent an absolute standardized risk score value of ±1.96 (corresponding to a *p*-value <0.05).

**Figure 7 ijms-22-04384-f007:**
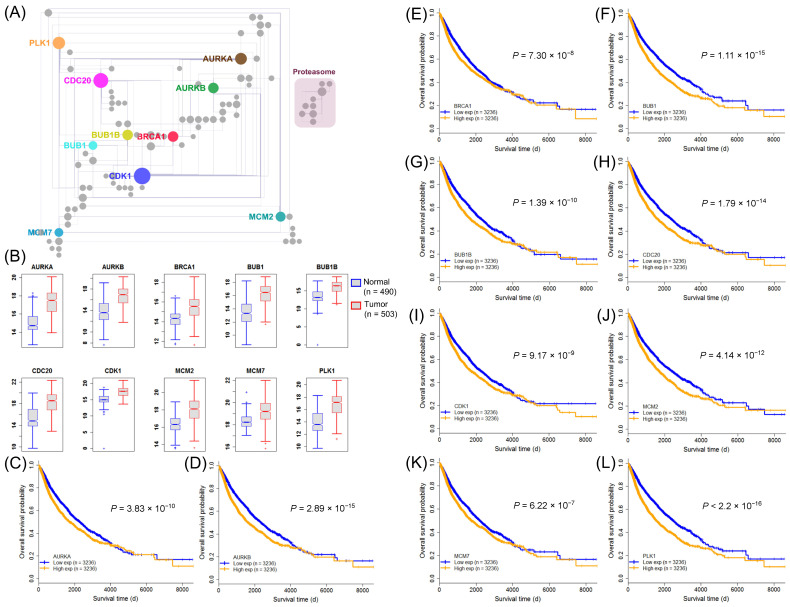
Clinically relevant genes in pan-cancer. (**A**) The human interactome of significantly co-expressed pan-cancer harmful genes (harmful proteins) in the MSigDB category of proliferation. Nodes represent genes and edges protein-protein interactions. Node size is proportional to the degree of the node. Nodes shown in color represent nodes with the 10 highest degrees. (**B**) Expression profile of clinically relevant genes in primary tumor and matched normal samples of pan-cancer. The RSEM normalized expression value is displayed in the log2 (x + 1) scale. (**C**–**L**) Survival estimates of overall survival in pan-cancer patients (n = 6584). Kaplan-Meier plots of low expression and high expression groups, based on the median expression. The *p*-values were obtained using a Mantel log-rank test.

## Data Availability

The list of identified SIGs and used clinical confounding factors are available online as Appendix A at www.mdpi.com.

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
