# Peer review of "Characterization of the Survival Influential Genes in Carcinogenesis"

_ijms, 2021, doi:10.3390/ijms22094384_

Round 1
Reviewer 1 Report
Introduction
1- The first sentence needs to be revised: —they are essential for tumor cells [1, 2]— needs to be removed.
2- The second sentence should be trimmed as this is very long at the current way.
3- "they are labor-intensive, time-consuming, and require a huge expenditure of money" is suggested to be changed as "they are capital and labor-intensive and time-consuming".
Results
The quality of the figures is poor. The details are not readable. Please replace them with higher-quality images.
Discussion
The authors need to discuss the biological relevance of the identified SIGs shown in Fig. 7, in terms of survival according to the previously published papers. In other words, the credibility of the findings needs to be highlighted by discussing the roles of these genes in the survival of tumor cells.
Reviewer 2 Report
This article is very interesting, but there are some points to revise before publishing.
- I think the last paragraph of “Introduction” included conclusion. Could you revise? Or, if this statement was based on previous your study, you should add references.
- The words of figure 4, 5, and 7 were too small to read. Could you revise?
- I would like to know how we use this results in clinical situation. What do you think about it?
